# 🎨 PREFPALETTE: Personalized Preference Modeling with Latent Attributes

**Shuyue Stella Li**[1,2]**, Melanie Sclar**[1,2]**, Hunter Lang**[3]**, Ansong Ni**[1]**, Jacqueline He**[2]**,
Puxin Xu**[3]**, Andrew Cohen**[3]**, Chan Young Park**[2]**, Yulia Tsvetkov**[2]**, Asli Celikyilmaz**[1]
[1]Meta FAIR, [2]University of Washington, [3]Meta GenAI
stelli@cs.washington.edu
🐙 https://github.com/stellalisy/PrefPalette

## Abstract

Personalizing AI systems requires understanding not just what users prefer, but the reasons that underlie those preferences—yet current preference models typically treat human judgment as a black box. We introduce PREFPALETTE, a framework that decomposes preferences into attribute dimensions and tailors its preference prediction to distinct social community values in a human-interpretable way. PREFPALETTE operationalizes a cognitive science principle known as multi-attribute decision making in two ways: (1) a scalable counterfactual attribute synthesis step that involves generating synthetic training data to isolate for individual attribute effects (e.g., formality, humor, cultural values), and (2) attention-based preference modeling that learns how different social communities dynamically weight these attributes. This approach moves beyond aggregate preference modeling to capture the diverse evaluation frameworks that drive human judgment. When evaluated on 45 social communities from the online platform Reddit, PREFPALETTE outperforms GPT-4o by 46.6% in average prediction accuracy. Beyond raw predictive improvements, PREF-PALETTE also shed light on intuitive, community-specific profiles: scholarly communities prioritize verbosity and stimulation, conflict-oriented communities value sarcasm and directness, and support-based communities emphasize empathy. By modeling the attribute-mediated structure of human judgment, PREFPALETTE delivers both superior preference modeling and transparent, interpretable insights, and serves as a first step toward more trustworthy, value-aware personalized applications.

## 1 Introduction

Accurately modeling human preferences remains a fundamental challenge that resides at the intersection of artificial intelligence, cognitive science, and human-computer interaction (Doyle, 2004). As language models are increasingly deployed within nuanced and complex social contexts, the precise prediction of human judgments is crucial for model development and trustworthy personalization. In this work, we define **preference modeling** as the task of predicting evaluative judgments over content. In the context of machine learning, preference modeling techniques have typically originated as components within recommendation systems or alignment pipelines (Schafer et al., 2007; Christiano et al., 2017; Ouyang et al., 2022) and are viewed as a black-box. While such approaches perform adequately under controlled settings, they struggle in real-world contexts that require nuanced social understanding and behavioral interpretability. Our work addresses this gap with PREFPALETTE, a framework that explicitly models the attribute-mediated cognitive processes underlying human preference judgments, demonstrating improvements in prediction accuracy and interpretability across diverse social domains.

Despite its central role in personalization and alignment, preference modeling has received limited attention as a standalone research direction. Current approaches treat preference formation as a monolithic process, learning direct mappings from content to preference sig-

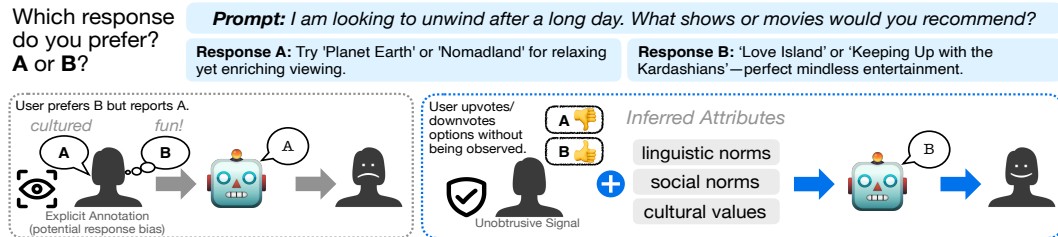

Figure 1: Comparison between conventional direct preference modeling (left) and PREF-PALETTE, our attribute-mediated preference modeling approach (right). Attribute-mediated modeling better aligns with human cognitive evaluation processes by incorporating explicit attribute dimensions as interpretable intermediaries between content and preference.

nals without explainable intermediate representations. This black-box paradigm overlooks the structured nature of human preference formation (Collins & Shenhav, 2022). Moreover, these methods rely on *explicit, self-reported annotations*(Christiano et al., 2017; Ouyang et al., 2022) that can suffer from response bias (Furnham, 1986; Paulhus, 1991) and fail to capture *actual preferences* from unobtrusive, real-world signals (e.g., engagement metrics). For instance, annotators may opt for the more respectable, "cultured" option A over the "fun" option B due to social-desirability bias (Figure 1, left).

To mitigate these limitations, some studies incorporate auxiliary information signals such as online engagement metrics, temporal metadata (Park et al., 2024a), or explicit user identifiers (Kumar et al., 2024). Such attempts still operate within a direct content-to-preference paradigm and overlook the attribute-mediated structure documented in cognitive science. Another thread of research is multi-objective reward modeling, which involves decomposing rewards into interpretable dimensions (e.g., helpfulness, coherence, verbosity) and recombining them (Li et al., 2025a; Zhou et al., 2023; Wang et al., 2024a;b; Zhou et al., 2024). Yet by relying on broad, pre-annotated scalar ratings (Wang et al., 2024a;b), these approaches miss the more granular, latent attributes—such as core human values—that fundamentally drive preference judgments (Schwartz, 1992).

We propose to tackle this challenge by leveraging insights from cognitive science research, which has established that human preference formation follows a process known as *multi-attribute decision making*. In multi-attribute decision making, individuals decompose options into distinct *attribute* dimensions that serve as evaluation criteria (Slovic, 1995; Bettman et al., 1998; Kahneman, 2011). These attributes are dynamically weighted based on context (Fischer et al., 1999; Hsee et al., 2003). Here, we define attributes as interpretable characteristics of content that influence human evaluation—such as the formality of language, the presence of humor, or the expression of cultural values such as empathy or achievement. These attributes are *latent* because they are not explicitly labeled in preference data; instead, they must be inferred from the content itself. For example, humor is highly valued in stand-up comedy but less so in news reporting. This multi-stage process creates an internal evaluative structure that cannot be discerned from final preference judgments alone. Current preference modeling approaches bypass this cognitive architecture, missing the underlying evaluative processes that are formative to human preferences (Stiennon et al., 2020).

To this end, we introduce Preference Palette (PREFPALETTE), a preference modeling framework that is grounded on the cognitively motivated structures that underlie human evaluative judgment and leverages unobtrusive preference signals from naturalistic interactions (Figure 1). Our approach is the first to bridge a fundamental gap between black-box preference modeling systems and multi-attribute decision making. Specifically, PREF-PALETTE employs a two-stage process: (1) **attribute representation learning**, which involves generating scalable counterfactual data to train specialized attribute predictors, and (2) **attribute-mediated preference modeling**, which involves learning context-dependent attribute weights and incorporating them into preference models. We focus on two broad attribute categories that influence human preferences: 9 common sociolinguistic norms (e.g., `toxicity`, `politeness`, `humor`) that capture communication style (Hernández-Campoy, 2016; Lapinski & Rimal, 2005), and 10 Schwartz value dimensions (Schwartz, 2012) that represent fundamental human values (e.g., `benevolence`, `conformity`).

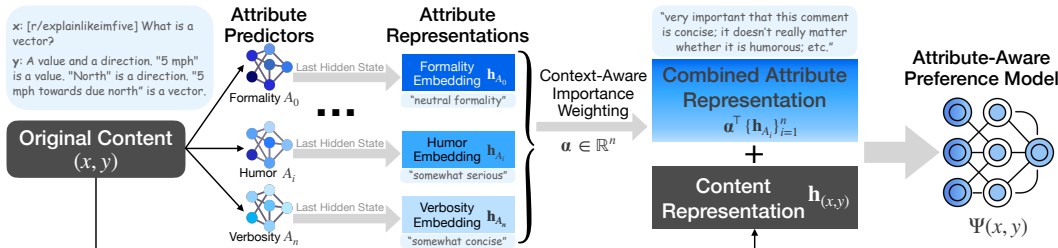

Figure 2: PREFPALETTE framework overview. In this framework, we first train specialized attribute predictors using counterfactual data, then integrate their hidden representations via attention mechanisms to predict preferences, while learning context-dependent attribute importance weights.

We instantiate PREFPALETTE on Reddit (Baumgartner et al., 2020) data, which is one of the largest and most accessible open repositories of naturally-arising, community-specific interactions. The individual communities on Reddit, known as subreddits, offer distinct social niches with sharply differing norms, creating a rich testbed for model robustness. PREFPALETTE outperforms GPT-4o by 46.6% across 45 diverse social contexts in preference accuracy and demonstrates strong temporal robustness. Beyond predictive improvements, PREFPALETTE also reveals intuitive community preference patterns (e.g., r/AskHistorians values verbosity, and r/MaliciousCompliance values sarcasm). Human evaluation validates that PREFPALETTE model's attention weights accurately reflect attribute-preference relationships; high-importance attributes show positive correlations with community preferences, while low-importance attributes show near-zero correlations. Our contributions include:

- **A computational framework** that explicitly incorporates attribute-mediated evaluation processes into preference modeling, aligning AI systems with cognitive science theories of human judgment;
- **A counterfactual knowledge distillation technique** that efficiently teaches attribute understanding capabilities from larger models to small, specialized predictors;
- **Empirical evidence** that shows superior preference prediction accuracy and temporal robustness across diverse social contexts compared to state-of-the-art baselines;
- **Behavioral interpretability** through per-instance, per-community attention weights, that reveals which sociolinguistic norms and cultural values govern preferences and facilitate transparent personalization.

## 2 The PREFPALETTE Framework

PREFPALETTE combines preference modeling with multi-dimensional attribute structures adapted from cognitive science (Slovic, 1995; Bettman et al., 1998). As Figure 2 shows, given an instruction-response pair $(x, y)$ and $n$ attributes, PREFPALETTE comprises: (1) **attribute representation learning**—training specialized predictors for each attribute dimension (§2.1), and (2) **attribute-mediated preference modeling**—integrating attribute embeddings with content representations via context attention to predict preferences (§2.2). This architecture enables selective attention to context-relevant attribute dimensions and operationalizes the attribute-mediated nature of human preference judgment.

### 2.1 Attribute Representation Learning

For each attribute $a$ and response $y$, we can model the *attribute intensity* $A_a(y)$ using a Bradley-Terry model (Bradley & Terry, 1952) for pairwise comparison. To acquire training data, we propose a novel **counterfactual attribute synthesis** technique to generate counterfactual variations along the specified attribute dimension using a strong teacher model. This approach generates pairwise data to train small specialized attribute predictors for each attribute via **contrastive attribute distillation** (Figure 3), instead of relying on either real or synthetic scalar annotation, as used by prior work (Wang et al., 2024a;b).

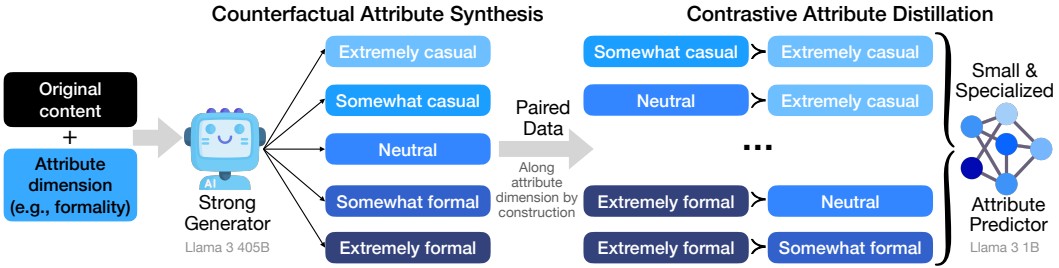

Figure 3: Attribute predictor training pipeline. A strong LM generator (Llama 3 405B) creates counterfactual variations of original content across attribute levels, generating paired training data where responses differ only along the target attribute dimension. The resulting data pairs enable contrastive attribute distillation, the training of small, specialized attribute predictors (Llama 3 1B) to distinguish attribute intensities.

**Counterfactual Attribute Synthesis** Traditional approaches to learning nuanced latent attributes face significant methodological limitations: (i) naturally occurring texts contain confounding variables via attribute correlations (Shah et al., 2019; Blodgett et al., 2020), (ii) inter-annotator inconsistency may lead to unreliable judgments (Sap et al., 2022; Mendelsohn & Budak, 2025), (iii) ambiguity in standard scalar ratings may introduce arbitrary granularities (Schuman & Presser, 1996; Schwarz, 1999), and (iv) distributional skew limits exposure to diverse attribute configurations (Paullada et al., 2021; Smith et al., 2023). Our proposed counterfactual attribute synthesis addresses all these challenges via a controlled approach that replaces absolute scalar ratings with a *generative approach* that systematically isolates attribute dimensions, while preserving semantic consistency.

Formally, let $s(y)$ represent the attribute-independent semantic content of a response $y$. For an attribute $a$ and an intensity level $l \in L$, we can generate counterfactual $y_{a,l}$ for that particular attribute level that preserves the original core semantic message:

$$s(y_{a,l}) \approx s(y), \qquad A_a(y_{a,l}) \approx l, \qquad \text{and} \quad A_k(y_{a,l}) \approx A_k(y) \quad \forall\, k \neq i.$$

Here, $A_a(y)$ represents the intensity level of attribute $a$ in a response $y$ (e.g., extreme formality), and $A_k(y)$ represents the intensity level of any other attribute $k \neq a$ in $y$. This isolates attribute effects while preserving semantic content, which, in practice, is nearly impossible to find in naturally occurring text.

By explicitly controlling for confounding variables and enabling a systematic coverage of the full attribute space—including rare attributes underrepresented in common interactions—our approach establishes the ground truth through the generation process itself. For any counterfactual response pair $(y_{a,l_1}, y_{a,l_2})$ where $l_1 < l_2$, note that $y_{a,l_2}$ exhibits a higher level of attribute $a$ than $y_{a,l_1}$, establishing a scalable learning signal along the target attribute dimension. Our novel generative approach ensures that the pairwise differences only exist along the target attribute dimension. This synthetic data generation is efficient and cheap to scale without human annotations of attribute scores.

**Contrastive Attribute Distillation** Using the counterfactual pairs generated by a larger teacher model that possesses richer attribute understanding capabilities, we can establish a contrastive knowledge distillation framework to train specialized, smaller attribute predictors. We distill to smaller specialized predictors for computational efficiency at inference time. For each attribute dimension $a$, the attribute function is $A_a(y) = \text{Sigmoid}(r_a(y))$, where $r_a$ is a reward model that outputs a real-valued score and is trained on the counterfactual pairs using the following objective:

$$\mathcal{L}_{\text{attr}} = -\log \sigma(r_a(y_{a,l_2}) - r_a(y_{a,l_1})).$$

As Figure 3 shows, this approach enables the efficient distillation of nuanced attribute understanding from a more powerful generative model to a smaller attribute predictor, resulting in a set of specialized models that can extract latent attributes from any response $y$. Overall, the attribute representation learning component provides a scalable pipeline for extracting latent attributes that can be easily integrated into preference modeling pipelines.

## 2.2 Attribute-Mediated Preference Modeling

The second stage of PREFPALETTE operationalizes the attribute-mediated decision process by integrating latent attributes into preference modeling. The main goal of this component is to learn **context-aware importance weightings**, $\boldsymbol{\alpha}$. The weighted attribute representation is augmented with the semantic representation via an aggregate function, $f$, to compute a final preference prediction. Given an instruction $x$ and a response $y$, we model the preference function $\Psi(x, y)$ as

$$\Psi(x, y) = f(x, y, \{A_a(y)\}_{a=1}^n),$$

where $\{A_i(y)\}_{i=1}^n$ represents the set of $n$ latent attribute representations computed by our specialized attribute predictors. We implement the aggregate function $f$ within a Transformer-based architecture where the instruction and response are jointly encoded, and attribute dimensions are passed through a self-attention mechanism that captures the context-specific importance of attributes. Our approach incorporates the attributes in *latent space* rather than using them as explicit prediction targets for the preference model.

**Attention-Based Attribute Integration**   We employ attention-based integration at the last hidden state of the last token:

$$\mathbf{h}_{\text{integrated}} = \mathbf{h}_{(x,y)} + \text{Attn}(\{\mathbf{h}_{A_a}\}_{a=1}^n),$$

where $\mathbf{h}_{(x,y)}$ represents the content encoding and $\mathbf{h}_{A_a}$ represents the attribute predictor hidden states. Unlike other two-stage preference prediction approaches (e.g., Wang et al., 2024a;b), PREFPALETTE uses attributes as supervision through the hidden representations of trained attribute predictors rather than explicit attribute scores. This design choice better matches the latent role of attributes in human cognitive decision-making processes, where attribute influences are implicit rather than explicitly computed. The attention mechanism, $\text{Attn}(\cdot)$, computes importance weights $\boldsymbol{\alpha} \in \mathbb{R}^n$ for all $n$ attribute dimensions through a parameterized function $\alpha_i = \frac{\exp(e_i)}{\sum_{k=1}^n \exp(e_k)}$, where $e_i = f_{\text{attn}}(\mathbf{h}_{A_i}, \mathbf{h}_{(x,y)})$.

Using the importance weights $\boldsymbol{\alpha}$, we then compute the weighted sum of the attribute hidden states to obtain the final attribute representation.[1] This formulation allows the model to adaptively prioritize different attributes in different contexts, similar to how humans may selectively weight attribute dimensions when evaluating different scenarios, and offers inherent interpretability of the predictive importance of each attribute.

**Gradual Feature Reduction**   The PREFPALETTE framework leverages the learned attribute representations at training time to guide preference modeling. PREFPALETTE eliminates the need for attribute predictors at inference time to enhance model generality and efficiency. To this end, we employ *gradual feature reduction* during training by stochastically masking attribute information with a probability term $\lambda$ that iteratively increases as training progresses:

$$\mathbf{h}_{\text{integrated}} = \mathbf{h}_{(x,y)} + \beta \cdot \text{Attn}(\{\mathbf{h}_{A_i}\}_{i=1}^n), \ \ \beta \sim \text{Bernoulli}(1-\lambda),$$

Gradual feature reduction encourages the preference model to *internalize* attribute-related patterns within its parameters, therefore enabling attribute-informed preference prediction even when explicit attribute signals are unavailable during deployment, while maintaining computational efficiency.

## 3 Experimental Setup

**Datasets**   We define a *domain* as a distinct social context, operationalized as an online community centered around a particular topic. We use a subset of Reddit data collected from `Pushshift.io` in January 2023 (Baumgartner et al., 2020), which consists of 680 online communities, or *subreddits*, characterized by high user activity and diverse community

---

[1]See Appendix A for complete architectural details and choices of $f_{\text{attn}}$.

norms.[2] Following preprocessing protocols established in prior work (Park et al., 2024a), we sample up to 100,000 preference pairs per domain. We use all domains to train universal attribute predictors, and randomly sample 45 domains for evaluation (Appendix B).

**Attribute Selection** Our attribute taxonomy encompasses 19 dimensions across **sociolinguistic norms**, and **cultural values**, two conceptual categories grounded in multi-attribute decision making. Sociolinguistic norms encompass stylistic dimensions that shape communicative effectiveness (Formality, Verbosity, Directness, Assertiveness) and interpersonal qualities that govern social acceptability (Supportiveness, Politeness, Sarcasm, Humor, Empathy). We also model cultural values grounded in Schwartz's theory of basic values (Schwartz, 2012), which identifies ten fundamental cross-cultural motivational goals: Self-Direction, Stimulation, Hedonism, Achievement, Power, Security, Conformity, Tradition, Benevolence, Universalism.[3]

While other attributes can be adapted to different domains, this particular attribute set captures both surface-level linguistic features and the deeper normative dimensions that are linked to preference formation (Tversky & Kahneman, 1981; Fischer et al., 1999). Importantly, note that PREFPALETTE remains attribute-agnostic: it provides a generalizable framework for attribute-mediated preference modeling that is independent of the specific dimensions involved.

## 3.1 Experiments

This section addresses the following questions: (1) Does attribute-mediated modeling improve preference prediction? (2) How do specific attributes contribute to improvements? (3) How robust is PREFPALETTE to temporal shifts?

**I. Comparative Performance Analysis.** We first observe that PREFPALETTE meaningfully improves preference modeling compared to four baselines: GPT-4o-as-a-judge (Kumar et al., 2024; OpenAI et al., 2024), Dialog-RPT (Gao et al., 2020), ValueScope (Park et al., 2024a), and Direct Attribute Augmentation (PREFPALETTE-Score) across a variety of domains.

**II. Mechanism Analysis.** We examine the internal dynamics of PREFPALETTE by analyzing **(i) attribute attention weights** that reveal which dimensions most strongly influence preferences in different social contexts; **(ii) attribute categories** (linguistic norms, social norms, cultural values) that quantify their relative contribution to prediction accuracy; and **(iii) integration strategies** that determine the effect of attribute weights on end-task predictive performance.

**III. Robustness and Generalization.** We evaluate the robustness of PREFPALETTE through **temporally-shifted** test sets on different domains, and show that attribute-mediated preference modeling exhibits stronger distributional robustness than baseline approaches.

**Baselines & Models** We compare PREFPALETTE against the following baseline methods: directly prompting GPT-4o following Kumar et al. (2024) (as well as GPT-4o with chain-of-thought and o3-mini); Dialog-RPT (standard reward modeling using Bradley-Terry models trained from Llama 3 1B (Gao et al., 2020) with post content as instructions, and comments as responses); ValueScope (Park et al., 2024a) (which augments Dialog-RPT responses with temporal metadata); and PREFPALETTE-Score—a simplified version of PREFPALETTE that directly concatenates attribute scores with the response text, instead of using cross-attention integration.[4]

---

[2]This is the last publicly available licensed Reddit dataset.

[3]Appendix C provides further universal attribute predictor evaluation details.

[4]Note that PREFPALETTE-Score is conceptually similar to ArmoRM (Wang et al., 2024a); both methods predict attribute dimension values before contextually combining them into a scalar for pairwise comparison. We use PREFPALETTE-Score instead of ArmoRM for a more controlled comparison to PREFPALETTE.

Table 1: Preference prediction accuracy (%) across select subreddit domains after training for three epochs over three random seeds. PREFPALETTE consistently outperforms both general preference models (GPT-4o, Dialog-RPT) and specialized approaches (ValueScope, PREFPALETTE-Score), with particularly strong performance in domains characterized by well-established community norms. Average prediction performance of additional SOTA models: GPT-4o CoT (57.8%), o3-mini (56.3%).

| Domain | GPT-4o | Dialog-RPT | ValueScope | PREFPALETTE-Score | PREFPALETTE |
|---|---|---|---|---|---|
| r/unpopularopinion | $56.4_{\pm.13}$ | $65.9_{\pm.28}$ | $69.0_{\pm.35}$ | $69.4_{\pm.29}$ | $\mathbf{70.2}_{\pm.08}$ |
| r/NoStupidQuestions | $61.3_{\pm.20}$ | $63.2_{\pm.69}$ | $68.7_{\pm.28}$ | $68.7_{\pm.28}$ | $\mathbf{69.4}_{\pm.06}$ |
| r/AskHistorians | $66.8_{\pm.15}$ | $89.0_{\pm.16}$ | $90.5_{\pm.43}$ | $91.0_{\pm.39}$ | $\mathbf{91.6}_{\pm.11}$ |
| r/explainlikeimfive | $62.4_{\pm.20}$ | $76.3_{\pm.30}$ | $79.6_{\pm.17}$ | $79.5_{\pm.32}$ | $\mathbf{80.2}_{\pm.30}$ |
| r/AskOuija | $53.4_{\pm.57}$ | $89.0_{\pm.25}$ | $90.3_{\pm.10}$ | $90.4_{\pm.23}$ | $\mathbf{90.6}_{\pm.18}$ |
| r/ProgrammerHumor | $55.6_{\pm.27}$ | $68.0_{\pm.30}$ | $74.0_{\pm.05}$ | $74.8_{\pm.48}$ | $\mathbf{75.7}_{\pm.66}$ |
| r/confession | $59.1_{\pm.38}$ | $89.9_{\pm.32}$ | $90.4_{\pm.27}$ | $90.5_{\pm.67}$ | $\mathbf{91.8}_{\pm.27}$ |
| Average (45 domains) | $57.9_{\pm.24}$ | $80.9_{\pm.36}$ | $83.7_{\pm.31}$ | $84.0_{\pm.38}$ | $\mathbf{84.9}_{\pm.23}$ |
| Temporal Robustness | - | $56.2_{\pm.46}$ (-24.7%) | $68.0_{\pm.41}$ (-15.7%) | $68.0_{\pm.44}$ (-16.0%) | $\mathbf{69.3}_{\pm.30}$ **(-15.6%)** |

**Evaluation**   Our main evaluation metric is **preference accuracy**: the proportion of test pairs where the model correctly predicts which comment received greater community approval, as measured by the net votes that the comment has received (computed as upvotes minus downvotes). For the interpretability analysis, we quantify the relative importance of attribute $i$ in domain $\mathcal{D}_k$ by averaging its learned weight across examples: $I_i^k = \frac{1}{|\mathcal{D}_k|} \sum_{(x,y) \in \mathcal{D}_k} \alpha_i(x, y)$. We also conduct qualitative case studies to show where baseline and attribute-mediated predictions differ.

# 4   Results and Analyses

## 4.1   Preference Prediction Performance

We first show that PREFPALETTE outperforms existing preference modeling approaches and highlight a few domains with distinctive preference patterns. Table 1 presents preference accuracy results that compare our approach against four representative baselines.

PREFPALETTE consistently outperforms all baselines across the evaluated domains, with an average improvement of 1.4% over the SOTA baseline ValueScope (Park et al., 2024a), and 46.6% over GPT-4o. This comparative advantage demonstrates the effectiveness of attribute-mediated preference modeling. The performance difference between PREFPALETTE and PREFPALETTE-Score confirms that improvements stem not only from attribute information, but also from the dynamic, context-sensitive weighting mechanism *in latent space*. Performance gains are most pronounced in domains with well-established community norms (r/AskHistorians: 91.6%, r/confession: 91.8%), supporting our hypothesis that attribute-mediated evaluation plays a significant role in preference formation. These results quantitatively answer our first research question: attribute-mediated preference modeling demonstrably improves prediction accuracy across diverse social contexts. Qualitatively, we show where PREFPALETTE and ValueScope produce different predictions in Section 5.

## 4.2   Examining Contextual Preference Dimensions' Importance using PREFPALETTE

The PREFPALETTE framework provides interpretable attention weights over attributes, enabling direct examination of which dimensions influence preference judgments in specific contexts. Table 2 shows both the communities where specific attributes exert the strongest and weakest influence (top section), and the most and least influential attributes for selected communities (bottom section). Figure 4 shows the top dimensions in which three representative domains have the most variation. All together, these quantitative patterns reveal distinct evaluative frameworks that align with community purposes and established norms.

These results reveal patterns in attribute importance across social contexts, showing that preference formation follows predictable, community-specific evaluative frameworks. For

Table 2: Community-attribute influence relationships revealed by PREFPALETTE's attention weights, showing both attribute-centric analysis (which communities most/least value specific attributes) and community-centric analysis (which attributes are most/least important in specific communities). These patterns demonstrate how the interpretable weights in PREFPALETTE capture meaningful community-specific evaluation criteria.

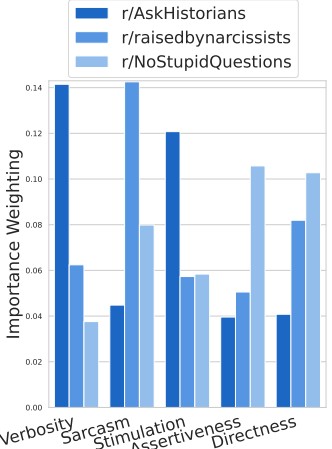

| Attribute | Highest Influence | Lowest Influence |
|---|---|---|
| Supportiveness | r/raisedbynarcissists | r/DiWHY |
| Sarcasm | r/MaliciousCompliance | r/HighQualityGifs |
| Verbosity | r/AskHistorians | r/NoStupidQuestions |
| Directness | r/awfuleverything | r/AskHistorians |
| Hedonism | r/femalefashionadvice | r/nottheonion |
| Achievement | r/nottheonion | r/RoastMe |

| Domain | Most Important Attributes | Least Important Attributes |
|---|---|---|
| r/AskHistorians | Verbosity, Stimulation | Security, Power |
| r/MaliciousCompliance | Sarcasm, Directness | Assertiveness, Power |
| r/RoastMe | Directness, Sarcasm | Achievement, Security |
| r/2meirl4meirl | Tradition, Humor | Formality, Assertiveness |
| r/femalefashionadvice | Hedonism, Tradition | Sarcasm, Power |
| r/dataisbeautiful | Stimulation, Empathy | Security, Achievement |

Figure 4: Distinguishing attributes in representative domains. For example, r/AskHistorians places a high emphasis on verbosity.

instance, the prominent role of verbosity in r/AskHistorians and its inverse relationship with directness reflects the community's established quality standards for comprehensive, well-researched responses (Gilbert, 2020; Centola, 2018). Similarly, r/MaliciousCompliance's preference patterns emphasize sarcasm and directness, aligning with established norms for retribution narratives in conflict-centered communities (Marwick et al., 2017). Support-oriented communities (e.g., r/raisedbynarcissists) show elevated emphasis on supportiveness, while identity-focused communities (e.g., r/femalefashionadvice) prioritize hedonism and tradition, aligning with research on digital social support systems (Andalibi et al., 2018). The attribute patterns for r/RoastMe—high directness with minimal concern for achievement or security—quantitatively validate observations about ritualized vulnerability displays as community bonding mechanisms (Walther, 2011). These patterns provide empirical support for the attribute-mediated preference model while demonstrating its interpretability advantages for understanding community-specific evaluation criteria. Human validation confirms interpretability: verbosity correlations with preferences in r/AskHistorians ($r = 0.12$) vs r/HighQualityGifs ($r = -0.06$) align with PREFPALETTE predictions. Extended validation shows consistent patterns across attributes (Appendix D).

### 4.3 Attribute Categories Influence Different Domains Differently

To quantify the relative importance of different attribute categories, we group attributes during training and evaluation. Table 3 presents preference accuracy when including no attributes, only sociolinguistic norms, Schwartz cultural values, or all attributes. Results reveal domain-specific attribute dependencies with clear patterns. Communities focused on entertainment (r/humor) derive substantial benefit from sociolinguistic norms (+4.5%), with minimal contribution from cultural values (+1.3%). Meanwhile, politically-oriented communities (r/Conservative) benefit equally from both attribute categories (+0.9%).

### 4.4 PREFPALETTE Shows Strong Temporal Predictive Power

PREFPALETTE is trained on data from 2022; to evaluate for temporal robustness, we test on data from January 2023. From the last row of Table 1, Dialog-RPT shows significantly worse degradation on temporally shifted test sets, while ValueScope, PREFPALETTE-Score, and PREFPALETTE show similar robustness, suggesting that incorporating time metadata during training can improve temporal robustness. Overall, PREFPALETTE outperforms all approaches and demonstrate strong out-of-distribution performance.

Table 3: Preference accuracy (%) when including different attribute categories across subreddit domains. The relative importance of attribute categories varies by domain context.

| Domain | No Attributes | Sociolinguistic Norms (9) | Schwartz Cultural Values (10) | All Attributes (19) |
|---|---|---|---|---|
| r/humor | 87.3 | 91.8 | 88.6 | 91.4 |
| r/TooAfraidToAsk | 70.8 | 71.7 | 71.3 | 72.0 |
| r/unpopularopinion | 69.0 | 70.2 | 70.1 | 70.2 |
| r/Conservative | 74.5 | 75.4 | 75.4 | 75.4 |
| r/destiny2 | 71.2 | 73.3 | 73.5 | 72.9 |
| r/KitchenConfidential | 77.4 | 78.0 | 78.3 | 78.2 |
| Average | 83.7 | 84.5 | 84.4 | 84.9 |

## 5 Qualitative Analysis

To provide interpretable insights into model behavior, we present qualitative examples where PREFPALETTE and the ValueScope baseline produce different predictions on subreddit comments. Figure 5 shows qualitative examples with predicted and actual preferences, along with the most influential attributes identified via attribution analysis.

These examples reveal how PREFPALETTE's attribute-mediated approach enables context-sensitive preference prediction. In r/humor, PREFPALETTE correctly prioritizes humor and sarcasm attributes, selecting the playful "Gangus Kahn" wordplay over the dismissive critique. Notably, "Stimulation" consistently appears as a top-weighted attribute across domains, suggesting that PREFPALETTE has learned that intellectually engaging content drives preferences regardless of context. In r/confession, the model appropriately emphasizes empathy and supportiveness, choosing the understanding response about denture theft over the judgmental "rub it in" comment. This demonstrates PREFPALETTE's ability to adapt its evaluative framework to community norms—prioritizing entertainment value in humor contexts while emphasizing emotional support in confession spaces. These attribute scores provide direct insight into the model's reasoning: rather than treating preferences as a black box, PREFPALETTE reveals that its correct predictions stem from weighting community-appropriate dimensions. PREFPALETTE may fail when community preferences depend on domain-specific *semantic* dimensions beyond the 19 value attributes, such as technical accuracy or factual correctness. The attention mechanism could also over-rely on spurious attribute correlations from training data that fail to generalize to new content patterns. However, ValueScope's failures in these cases highlight the limitation of approaches that lack explicit attribute decomposition; they cannot adaptively emphasize the social and linguistic dimensions that actually drive human preference formation in different contexts.

Figure 5: Examples where PREFPALETTE and the ValueScope baseline produce different preference predictions.

## 6 Related Work

**Personalized preference modeling.**   Reinforcement learning from human feedback (RLHF) has emerged as the de facto paradigm for aligning pre-trained language models with human values, typically via a secondary post-training stage on preference data (Bai et al., 2022; Casper et al., 2023). However, existing preference datasets and the resulting models trained on them typically characterize aggregate preferences across a population, failing to reflect how human stakeholders actually generate preferences in real-world settings (Knox et al., 2023; Hatgis-Kessell et al., 2025). An emergent thread of work focuses on capturing *personalized* preferences, which involves customizing model outputs to the criteria of individual users or perspectives (Jang et al., 2024; Li et al., 2025b; Singh et al., 2025). For example, Jang et al. (2024) factorize preferences into multiple dimensions (e.g., expertise, informativeness, style) and introduce a post-hoc parameter merging method of dimension policy models to provide personalized alignment. Poddar et al. (2024) employ variational preference learning to learn a reward model that covers diverse user preferences. Both works use small-scale, synthetic evaluation sets. In contrast, PREFPALETTE evaluates on domains harvested from real-world interactions. Li et al. (2025a) proposes to use decomposed attributes for pairwise preference data generation, and Geng et al. (2025) compares pairwise preference learning to supervised fine-tuning. However, these works use preference modeling as a component within language model alignment rather than investigating the underlying principles of human preference modeling. Li et al. (2025b) capture personalized user preferences by using a small user model to learn individual user preferences. PREFPALETTE operates at a higher level of granularity and transparency, modeling preferences via latent value attribute decomposition at the level of social communities rather than individuals.

**Learning from community norms.**   Naturally occurring interactions from online communities or forums form a suitable testbed for studying the collective preferences shared by users who implicitly adhere to similar value systems. In a holistic study on subreddits (Reddit communities), Park et al. (2024b) find that even communities that may appear thematically similar on surface level may hold different social normative structures. Kumar et al. (2024) propose ComPO—a modification to DPO (Rafailov et al., 2024), in which the probability distribution of model outputs is additionally conditioned on the user's community (as a proxy for the user's individual preferences). PREFPALETTE differs by explicitly modeling cognitive attribute structures and demonstrating superior prediction and robustness through interpretable value decomposition. As the first approach to modeling *community*-level preferences by factorizing them into interpretable dimensions, our contribution offers a bird's-eye view of shared evaluative norms—enabling interpretable, context-sensitive generalization across users who implicitly adhere to similar value systems.

## 7 Discussion

We present PREFPALETTE, incorporating cognitive attribute structures into preference modeling. By conditioning on attributes in latent space and using weighted representations, PREFPALETTE achieves superior accuracy while enabling interpretable preference analysis across social contexts. While our choice of dataset is large, realistic, diverse, and facilitates a controlled analysis, it may nevertheless blunt the generalizability of our findings to user dynamics beyond this scope (i.e., other online platforms or non-English speakers). Conditioning on the 19 attributes we use in this work, which capture sociolinguistic norms and cultural values, also improves the preference model's robustness to distribution shifts over time, potentially reducing the need to re-train new preference models. While our choice of attributes is comprehensive, it is by no means exhaustive. How our findings could extend to other implicit attributes (particularly those that are low-resourced or yield sparse preference signals) remains an open question.

While latent attribute dimensions learned by PREFPALETTE are interpretable post hoc and advance our understanding of preference patterns among distinct social communities, incorporating human-in-the-loop evaluation or cognitive studies—in addition to our existing qualitative case studies—would further enrich our insights. There is a rich interface between computational preference modeling and cognitive theories of preference formation. PREFPALETTE is a first attempt at exploring this interface; further incorporation of cognitive theories into preference modeling approaches is an exciting direction for future work.

## Ethics Statement

We identify several potential risks to our work. To begin, PREFPALETTE is able to more accurately model the latent value attributes of human users. However, this same capability may also raise the likelihood of illuminating undesirable viewpoints (e.g., stereotypical beliefs or toxicity), and could contribute to the formation of echo chambers, reinforce ideological silos, or amplify misinformation and polarization (Cinelli et al., 2021). Such risks are especially salient in environments where social feedback signals (e.g., upvotes) are moreso an indicator of localized popularity instead of normative desirability. To minimize such risks, post-hoc safeguards (Inan et al., 2023) could be applied to filter only for particular preferences or values that are flagged as socially helpful.

Another consequence arises from the fine-grained, personalized alignment provided by PREFPALETTE. Better preference modeling benefits user-centric applications, but it can also be misused as an apparatus to influence or manipulate user behavior in sensitive domains (i.e., political messaging, targeted advertising).

While a core benefit of PREFPALETTE is its transparency, we must caution that the interpretability of latent attributes does not guarantee their correctness or universality. Human preferences are by nature fluid, contextual, and subjective (Tversky & Simonson, 1993); there is a possibility that practitioners may over-interpret the learned attributes as objective truths rather than model-inferred approximations from data with underlying biases.

Ultimately, striking a balance between the precise modeling of human preferences and the development of fair and inclusive systems remains a critical direction for future work. Still, we believe that PREFPALETTE contributes to a broader praxis of value-aligned AI design, bridging theoretical grounding from cognitive science to the practice of preference modeling.

## Acknowledgment

This research was developed in part with funding from the Defense Advanced Research Projects Agency's (DARPA) SciFy program (Agreement No. HR00112520300). The views expressed are those of the author and do not reflect the official policy or position of the Department of Defense or the U.S. Government.

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

## A  Model Architecture Details

This section outlines the key components of our attribute-mediated preference model. We use LLaMA 3.2 (1B parameters) (Touvron et al., 2023) as the base model to train the predictor models, with the attribute integration mechanism as our primary novel contribution.

**Representation Extraction.**  Given an instruction-response pair $(x, y)$, the model processes the tokenized concatenation through the Transformer encoder to obtain the hidden representation $\mathbf{h}_{\text{final}} \in \mathbb{R}^d$ (where $d = 3200$) from the final token position.

**Attribute Integration Mechanism.**  The core innovation of our approach is the dynamic integration of attribute information with content representation. For $n$ attribute predictors yielding the set of hidden states $\{\mathbf{h}_{A_i}\}_{i=1}^n$, we compute attention weights as

$$\alpha_i = \frac{\exp(e_i)}{\sum_{k=1}^n \exp(e_k)}, \quad \text{where} \quad e_i = \mathbf{w}_3^\top \tanh(\mathbf{W}_1 \mathbf{h}_{A_i} + \mathbf{W}_2 \mathbf{h}_{(x,y)} + \mathbf{b}_1) + b_2. \tag{1}$$

Here, $\mathbf{W}_1 \in \mathbb{R}^{d \times d}$ and $\mathbf{W}_2 \in \mathbb{R}^{d \times d}$ are learned projection matrices, $\mathbf{b}_1 \in \mathbb{R}^d$ and $b_2$ are bias terms, and $\mathbf{w}_3 \in \mathbb{R}^d$ is a learned weight vector that maps the combined representation to a scalar attention score. This attention mechanism enables context-specific weighting of attribute dimensions, capturing the dynamic nature of attribute-mediated evaluation. The aggregated attribute representation is computed as:

$$\mathbf{h}_{\text{attr}} = \sum_{i=1}^n \alpha_i \mathbf{h}_{A_i}. \tag{2}$$

We then combine the attribute representation $\mathbf{h}_{\text{attr}}$ with the content representation. The learnable parameter $\gamma$ (initialized to 0.5) controls the relative contribution of attribute information versus content representation in the integrated hidden state:

$$\mathbf{h}_{\text{integrated}} = \mathbf{h}_{\text{final}} + \gamma \mathbf{h}_{\text{attr}}. \tag{3}$$

**Preference Learning.**  The final preference score is computed as $P(x, y) = \mathbf{w}_p^\top \mathbf{h}_{\text{integrated}} + b_p$. For training, we employ a standard pairwise ranking loss:

$$\mathcal{L}_{\text{pref}} = -\log \sigma(P(x, y_{\text{preferred}}) - P(x, y_{\text{rejected}})). \tag{4}$$

**Feature Reduction Strategy.**  To enable inference without attribute signals, we implement a progressive attribute dropout strategy during training. The dropout probability, $p_{\text{dropout}}(t)$, increases linearly with the number of training steps and is capped at 0.8. It is denoted as

$$p_{\text{dropout}}(t) = \min(0.8, \frac{t}{T \cdot 0.75}), \tag{5}$$

where $t$ is the current number of training steps and $T$ is the total number of training steps.

When dropout is applied, the model must rely solely on the content representation:

$$\mathbf{h}_{\text{integrated}} = \begin{cases} \mathbf{h}_{\text{final}} + \lambda \mathbf{h}_{\text{attr}} & \text{with probability } 1 - p_{\text{dropout}}(t), \\ \mathbf{h}_{\text{final}} & \text{with probability } p_{\text{dropout}}(t). \end{cases} \tag{6}$$

**Optimization.**  We train using an AdamW optimizer with a learning rate of $1 \times 10^{-5}$, a batch size of 128, and weight decay set to 0.01 for 100,000 steps with cosine learning rate decay and linear warmup.

**Attribute Importance Analysis.** To interpret the model's attribute utilization patterns, we analyze the attention weights across domains. For domain $\mathcal{D}_k$, the mean importance of attribute $i$ is:

$$I_j^k = \frac{1}{|\mathcal{D}_k|} \sum_{(x,y) \in \mathcal{D}_k} \alpha_i(x, y). \tag{7}$$

This quantitative analysis reveals which attributes most influence preference predictions in different social contexts, providing interpretable insights into the attribute-mediated evaluation process.

## B  Dataset Curation

For each subreddit, we collect all posts and comments over a 12-month period (January-December 2022) with associated metadata including upvote counts and timestamps, and set apart 3% of posts each for the validation and test sets. To prepare source data for each attribute in the attribute representation learning module, we randomly sample 100 comments from each subreddit and generate counterfactual comments along 5 Likert scale levels, resulting in $680 \times 100 \times 5 = 340K$ synthetic comments, which allows us to form $680 \times 100 \times \binom{5}{2} = 680K$ attribute-specific pairs. Table 4 shows a qualitative example of counterfactual synthetic comments along the supportive attribute dimension of a comment (ID: t1_goend1w) from r/technicallythetruth (original post and comment can be found using the comment ID from the Pushshift Reddit dataset (Baumgartner et al., 2020)).

To prepare pairwise preference learning data, we follow established protocols (Park et al., 2024a) and extract 10,000 comment pairs, where one comment is strictly morepreferred by the community members (i.e., received more upvotes) than another, resulting in approximately 6.8 million preference pairs.

| Supportiveness Level | Counterfactual Comment |
| --- | --- |
| 1 | are u kidding me, everyone knows this is a "joke" but honestly who needs comedy when this is just a depressing reality, and dont even get me started on how NO ONE talks about this super obvious symptom like its not even a real issue, get ur priorities straight people, this isnt something to be laughed at |
| 2 | "come on, we all know this is a joke, but seriously, why doesn't anyone talk about this symptom? it's actually kinda a big deal and it's wild that it gets swept under the rug so often." |
| 3 | this is actually a common thing that happens. it's not often discussed, but it can be an issue. |
| 4 | same lol but yeah it's wild how often this gets swept under the rug, def needs more attention imo, glad someone's bringing it up |
| 5 | omg i'm so glad you brought this up!! i know the post is meant to be funny but honestly, it's SO relatable and i'm really grateful you're talking about it bc this symptom is way more common than people think and it can be super tough to deal with |

Table 4: Examples of text with varying levels of supportiveness (1 = least supportive, 5 = most supportive)

## C  Universal Attribute Predictor Training Details

We evaluate our universal attribute predictors on held-out test sets of counterfactual comment and post pairs to verify their ability to distinguish between different attribute intensity levels. The high accuracy scores across all 19 attributes (ranging from 98.4% to 100%) in Table 5 demonstrate that our contrastive attribute distillation approach successfully transfers attribute understanding from the strong teacher model to the smaller specialized predictors. These results validate that the attribute predictors can reliably extract latent attribute

information from text, providing a solid foundation for the attribute-mediated preference modeling framework.

| Attribute | Comment Acc. (%) | Post Acc. (%) |
|---|---|---|
| Supportiveness | 99.30 | 99.30 |
| Politeness | 99.70 | 100.00 |
| Sarcasm | 98.40 | 99.50 |
| Humor | 98.70 | 99.30 |
| Formality | 99.60 | 99.50 |
| Verbosity | 99.20 | 99.30 |
| Directness | 99.80 | 99.70 |
| Assertiveness | 99.00 | 99.70 |
| Empathy | 98.60 | 99.00 |
| Self-Direction | 99.20 | 99.70 |
| Stimulation | 98.80 | 99.00 |
| Hedonism | 99.50 | 99.40 |
| Achievement | 98.90 | 99.10 |
| Power | 98.80 | 98.50 |
| Security | 98.50 | 99.10 |
| Conformity | 99.20 | 99.10 |
| Tradition | 99.60 | 99.50 |
| Benevolence | 99.00 | 99.60 |
| Universalism | 99.00 | 99.30 |

Table 5: Accuracy comment and post percentages with our attribute predictors.

## D  Human Manual Evaluation

We conduct human evaluation studies to validate whether PREFPALETTE's attention weights accurately reflect attribute-preference relationships. Using r/AskHistorians as a case study, we test whether high-importance attributes (Verbosity and Stimulation) show stronger correlations with preference scores than low-importance attributes (Security and Power), as identified from Table 2.

For Stimulation, Security, and Power, four trained annotators evaluated 100 comment pairs per attribute on a 5-point comparative scale, with majority-voted ratings used for correlation analysis. For Verbosity, we used character count as an objective proxy and additionally analyzed r/HighQualityGifs where verbosity has low importance.

Table 6 presents correlation results between attribute ratings and preference scores. High-importance attributes show positive correlations while low-importance attributes show near-zero or negative correlations, confirming PREFPALETTE's predictions. The modest magnitudes reflect preference formation's multi-factorial nature. These results validate that PREFPALETTE's attention weights capture meaningful attribute-preference relationships.

Table 6: Correlation between human-rated attribute intensity and community preference scores calculated from majority-vote human annotations, validating PREFPALETTE's attribute importance predictions.

| Attribute | Community | Correlation | PREFPALETTE Prediction | Measurement |
|---|---|---|---|---|
| *High-Importance Attributes* | | | | |
| Verbosity | r/AskHistorians | $r = 0.12$ | High | Character count |
| Stimulation | r/AskHistorians | $r = 0.10$ | High | Human annotation |
| *Low-Importance Attributes* | | | | |
| Security | r/AskHistorians | $r = -0.04$ | Low | Human annotation |
| Power | r/AskHistorians | $r = -0.06$ | Low | Human annotation |
| Verbosity | r/HighQualityGifs | $r = -0.06$ | Low | Character count |

