# OpenReview forum: "PrefPalette: Personalized Preference Modeling with Latent Attributes"
_colmweb.org/COLM/2025/Conference — COLM 2025_

### Official Review · Reviewer_do5a · 2025-05-05

**Rating:** 7
**Confidence:** 4
**Ethics Flag:** 1

**Summary:**

This work tackles preference modeling, i.e. predicting evaluative judgement over a given content, introducing a framework dubbed LAVA, for LAtent Value Attribute, modeling the cognitive process, with good results in accuracy and interpretability in their experiments. LAVA works by training predictors for the attributes, then used to produce attribute embeddings which are integrated with the content representation to predict a preference. The accuracy of LAVA when predicting the preferences for a Reddit dataset bests their baselines, with an impressive margin for the generic GPT-4o. Admittedly LAVA barely does better than the SoTA baseline but it has interpretable weightings, well in line with the specific subreddits.
Preference modeling is a very relevant subfield. The intuition of relying on latent value attributes makes a lot of sense as it can indeed explain many of the preferences, translating in the good results shown here.
The paper itself is well written and a nice read. The work seems sound, if limited to a single dataset, which might question the applicability of the approach to other preferences, where the picked attributes might not translate to similar results –the fact that it works well for multiple subreddits does not mean it would transfer beyond the very specific world of Reddit.

**Questions To Authors:**

- how would you approach a completely new domain, beyond Reddit's culture? I'm curious about the dimensions: do you think they would transfer well?

**Reasons To Accept:**

This is a nice, theoretically grounded, work showing good interpretable results:
- grounding in cognitive science: it is intuitive but this grounding does help.
- good performances: when predicting preferences, especially over GPT-4o used as a judge
- interpretability: by construction, and well in line with the expected values of the subdomains
- resistance to temporal shift: although the experiment is limited in that regard

**Reasons To Reject:**

It's not clear if the results would extend beyond Reddit, at least without more work on the attributes:
- limited experiments with a focus on a single dataset, in the artificial world of Reddit.
- attribute selection: it's not clear how the dimensions were picked not if they would apply in different settings.
- reliance on synthetic data: it's not clear how the counterfactual data are able to capture the needed nuances.
- resistance to shift: there is some robustness but how much? As mentioned above, the experiment is very limited.

---

> ### Author Response · Authors · 2025-06-03
>
> We thank the reviewer for the positive evaluation and insightful feedback. We are pleased that the reviewer found our work "theoretically grounded" with "good interpretable results" and appreciated the "grounding in cognitive science," "good performances," and "interpretability by construction." We address their concerns about generalizability below.
>
> ***1. Attribute Selection and Theoretical Grounding***
>
> > "attribute selection: it's not clear how the dimensions were picked not if they would apply in different settings."
>
> Our LAVA framework is attribute-agnostic, and can theoretically work with any set of meaningful attributes. Moreover, the selection of our specific attributes is not Reddit-specific but grounded in established cognitive science theories.
>
> Our attribute selection is theory-driven rather than data-driven or arbitrarily chosen: the 9 sociolinguistic norms capture universal communication patterns documented across cultures and contexts (formality, politeness, humor, etc.), while the 10 Schwartz values represent fundamental human motivational goals that have been validated across 82 countries and diverse cultural settings, and has been used in prior work [1,2].
>
> This theoretical foundation suggests strong transferability to general-purpose tasks beyond Reddit community data, while these selection of attributes focus on enhancing the social intelligence of the models, the attributes can always be customized to the task itself.
>
>
> ***2. Clarification on Counterfactual Data Quality***
>
> > "reliance on synthetic data: it's not clear how the counterfactual data are able to capture the needed nuances."
>
> Our counterfactual generation uses the exact same prompt as Park et al., 2024 [3] (Section 4.2), whose authors performed extensive human annotation to validate the quality of synthetic variations. We did not repeat this expensive annotation process due to budget constraints.
> Additionally, the strong empirical results and intuitive interpretability conclusions across diverse subreddits suggest our synthetic data effectively captures the necessary nuances for preference modeling.
>
>
> ***3. Robustness to Shifts***
>
> > "It's not clear if the results would extend beyond Reddit, at least without more work on the attributes. I'm curious about the dimensions: do you think they would transfer well?"
>
> > "resistance to shift: there is some robustness but how much? As mentioned above, the experiment is very limited."
>
> These are excellent questions about the broader applicability and robustness of our approach. Reddit is one of the largest and most accessible open repositories of naturally-arising, community-specific interactions on the internet. The openness and ease of data collection, along with the diversity and scale of social interactions are key reasons why we primarily consider this data set. We showed that our models are robust against temporal shifts, demonstrating superior stability compared to baselines (15.6% drop vs. 16-24.7% for others).
>
> We agree with the reviewer that testing on multiple different subreddits does not necessarily imply generalizability beyond Reddit communities. However, we want to highlight the substantial norm differences among subreddits as shown in prior work, as well as the significant temporal shift we evaluated as supported by prior work [4].
>
> Regarding broader generalizability, we believe our cognitive attributes should transfer well to other domains because the attribute selection was not Reddit-based. To empirically validate this, we have begun training and evaluating LaVA on Amazon product review datasets–a smaller dataset with more limited diversity. Due to rebuttal time constraints, training is ongoing but results are not yet complete. We will update with transfer results as soon as they are available.
>
> We appreciate the reviewer's positive assessment and believe our ongoing cross-domain evaluation will address the primary concerns about generalizability while maintaining the theoretical strengths the reviewer praised.
>
>
> [1] Huang, S., Durmus, E., McCain, M., Handa, K., Tamkin, A., Hong, J., ... & Ganguli, D. (2025). Values in the wild: Discovering and analyzing values in real-world language model interactions. arXiv preprint arXiv:2504.15236.
>
> [2] Moore, J., Deshpande, T., & Yang, D. (2024). Are Large Language Models Consistent over Value-laden Questions?. arXiv preprint arXiv:2407.02996.
>
> [3] Chan Young Park, Shuyue Stella Li, Hayoung Jung, Svitlana Volkova, Tanushree Mitra, 556 David Jurgens, and Yulia Tsvetkov. Valuescope: Unveiling implicit norms and values via 557 return potential model of social interactions, 2024a. URL https://arxiv.org/abs/2407. 558 02472.
>
> [4] Cristian Danescu-Niculescu-Mizil, Robert West, Dan Jurafsky, Jure Leskovec, and Christopher Potts. 2013. No country for old members: user lifecycle and linguistic change in online communities. Proceedings of the 22nd international conference on World Wide Web.

---

> > ### Comment · Reviewer_do5a · 2025-06-07
> >
> > thanks for this long answer, I am convinced by the quality of this work but I do appreciate the extra explanations —subreddits might indeed have "substantial norm differences", thanks for the reference. I noted the intuition about "broader generalizability" and the extra evaluation on the Amazon dataset, which should address my concern.

---

### Official Review · Reviewer_3hQX · 2025-05-12

**Rating:** 5
**Confidence:** 4
**Ethics Flag:** 1

**Summary:**

This paper introduces LAVA, a framework including both attribute representation learning and preference modeling with the aid of attribute information. The authors seem to primarily focus on LLMs' preference prediction accuracy and robustness while achieve more interpretability based on attributes from cognitive science theories.

This paper is not easy to follow to understand its positioning statement. In Section 1 the authors compare two paradigms: one with ``self-reported preference`` and the other with ``actual preferences``. I can understand response biases but not sure how the proposed approach and experiments address response biases in preference data. Figure 1 is presented around there but is never used and explained in the main text. From Figure 1 it seems the ``self-reported preference`` paradigm like [1] asks users to annotate attributes like ``funny`` and ``cultured`` to responses but this is not my understanding of [1].

This paper does not provide a mathematical formulation of the overall setup and it is difficult to infer that the main objective is to improve preference prediction accuracy. For attribute representation learning, the proposed framework first generates attribute training data with counterfactual variation and then trains attribute predictors with Learning to Rank [2]. Their core contribution is to integrate latent attribute representation from Section 2.1 into preference modeling through attention mechanism. On the other hand, it is known that multi-task learning can improve performance of each task and people have some success in LLMs [3]. The authors should compare with multi-task learning.

I do not get the intuition behind Gradual Feature Reduction. The paper says it is ``to eliminate the need for attribute predictors`` but we still need their last hidden state. I can see we want to deemphasize the influence of attribute predictors as the goal is still preference prediction but this annealing process is not principled for interpretability. Alternatively, one can apply the TCAV [4] methodology to preference prediction with attributes from cognitive science theories being the concepts. It would be interesting to compare the LAVA framework with the TCAV methodology. With TCAV, we can mathematically establish an attribute being a deciding factor in predicting preference. However, LAVA doesn't seem to have this property because it needs $h_{(x,y)}$ below L168 to help preference prediction.

The  experimental design of testing prediction accuracy and robustness is reasonable. It sounds like doing in-context learning (DAA) can already beat SoTA. The gains in robustness are not promising which hurts the argument of the paper. The experimental results related to interpretability like Table 2 are not validated by human evaluation and it is unclear if the results are consistent with actual human decision making.

[1] Training language models to follow instructions with human feedback (Long Ouyang et al., 2022)

[2] Learning to rank using gradient descent (C.J.C. Burge et al., 2005)

[3] Cappy: Outperforming and Boosting Large Multi-Task LMs with a Small Scorer (B. Tan et al., 2023)

[4] Interpretability Beyond Feature Attribution: Quantitative Testing with Concept Activation Vectors (TCAV) (B. Kim et al., 2018)

**Questions To Authors:**

* L148: It is inconsistent that we use $\sigma$ in the loss function of attribute predictor $- \log \sigma(r_i(y_i,l_2 ) − r_i(y_i,l_1 ))$ given we just say $Sigmoid(r_i(y))$ in L147.
* L172: "Using the $\alpha_i$’s we then" ->  $\alpha_i$’s what?
* L172: I couldn't find $h(x,y)$ in Appendix A so don't really understand $f_{attn}(h_{A_i} , h(x,y))$.
* L192: "we sample up to 100K preference" but "10,000 comment pairs" in L663.
* L305: I don't know why ``LAVA is the only approach that outperforms GPT-4o in zero-shot settings, demonstrating strong out-of-distribution performance.`` as both ValueScope and DAA also outperform GPT-4o.
* L337: Can authors elaborate their experimental setup and evidence "In a controlled comparison against ComPO" as it does not seem to be described in the paper?

**Reasons To Accept:**

* The addition of attributes to understand human preference modeling.
* Generating attribute training data with counterfactual variation is good.
* The direction of improving temporal robustness with attributes.

**Reasons To Reject:**

* A lack of mathematical formulation of the overall problem. Need to improve the positioning statement.
* A lack of understanding of preference prediction improvement and comparison with multi-task learning.
* The proposed framework is not principled for interpretability. Need to compare with TCAV.
* The results, especially in robustness, are not promising enough.
* No human study related to interpretability.

---

> ### Author Response · Authors · 2025-06-03
>
> We thank the reviewer for providing the thoughtful review. We appreciate the recognition of several strengths in our work, including the unique conceptual advantage of “**achieving more interpretability based on attributes from cognitive science theories**” to "**understand human preference modeling**," "**counterfactual variation data generation pipeline is good**," and the empirical advantage of "**improving temporal robustness with attributes**." We address the main concerns below.
>
> ***1. Mathematical Formulation and Problem Setup***
>
> > "This paper does not provide a mathematical formulation of the overall setup."
>
> Thank you for bringing up this point. We have included detailed mathematical formulations in Appendix A to support the problem set-up in  lines 159-161. But following the Reviewer’s suggestion, we will add the following formal definitions early in the paper:
>
> **Problem Setup**: Given an instruction-response pair $(x, y)$ and a set of $n$ attributes $\mathcal{A} = {A_1, A_2, \ldots, A_n}$, we aim to predict human preference judgments $\Psi(x, y)$ where $P$ represents the true underlying preference distribution.
>
> **Traditional Approach**: Existing methods learn a direct mapping: $\Psi_{\text{direct}}(x, y) = f_{\theta}(x, y)$
>
> **Our Attribute-Mediated Approach**: We decompose preference prediction through latent attributes: $\Psi_{\text{LAVA}}(x, y) = g_{\phi}\left(h(x,y), {A_i(y)}_{i=1}^n\right)$
>
> **Objective**: Minimize the gap between predicted and true preferences: $\mathcal{L} = \mathbb{E}{(x,y) \sim \mathcal{D}} \left[ \ell(\Psi*(x,y), \Psi_{\text{LAVA}}(x,y)) \right]$
>
> ***2. Response Bias and Figure 1***
>
> > "I can understand response biases but not sure how the proposed approach and experiments address response biases in preference data. Figure 1 is presented around there but is never used and explained in the main text."
>
> To clarify the approach in RLHF [1]: when option A is considered "cultured" and option B is considered "fun," annotators may feel pressured to pick the one that's more highly regarded due to response bias [2]. Our approach addresses this by using **unobtrusive preference signals** from natural human-human interactions rather than explicit annotation tasks. Since upvote/downvote behaviors on Reddit are private and not visible to other users, this reduces social desirability bias compared to deliberate annotation scenarios. We will add this clarification in Section 1.
>
>
> ***3. Human Evaluation of Interpretability***
>
> > "The experimental results related to interpretability like Table 2 are not validated by human evaluation and it is unclear if the results are consistent with actual human decision making."
>
> Although interpretability is not the primary focus of this work, following the Reviewer’s suggestion we have designed a human annotation scheme to validate our interpretability findings. Given an attribute and comments from two subreddits where our model predicts different attribute importance levels, we ask annotators to rate the attribute intensity in each comment. We then test whether the correlation between human ratings and preference labels is stronger in the high-importance subreddit versus the low-importance one.
>
> We did a pilot study on the verbosity attribute, using character count as a proxy for human-annotated attribute intensity. Verbosity is a high-importance attribute for r/AskHistorian—a community about historical topics  with detailed and well-researched answers—and a low-importance attribute for r/HighQualityGIFs—a community where the main medium is high-quality animated GIFs.
>
> The Pearson’s correlation coefficients between verbosity and preference scores in r/AskHistorians is **r=0.12** versus **r=-0.06** in r/HighQualityGifs, **aligning with LAVA's predictions**. We expect the correlation between any single attribute and the score to be low; only the relative difference matters here. Full annotations for other attributes are deployed and complete results will be in the revision.
>
> ***4. Multi-task Learning Comparison***
>
> > "It is known that multi-task learning can improve performance of each task and people have some success in LLMs. The authors should compare it with multi-task learning."
>
> LaVA already incorporates an implicit multi-task structure via shared encoders and attention-based aggregation. Methodologically, we learn to dynamically weight attribute dimensions for a single preference task rather than learning multiple separate tasks. Training a fully-fledged multi-task baseline would require an extensive hyperparameter search that is beyond our current scope, so we leave this direction for future work.
>
> [1] Training language models to follow instructions with human feedback (Long Ouyang et al., 2022)
>
> [2] Antin, J., & Shaw, A. (2012, May). Social desirability bias and self-reports of motivation: a study of amazon mechanical turk in the US and India. In Proceedings of the SIGCHI Conference on human factors in computing systems (pp. 2925-2934).

---

> > ### Author Response · Authors · 2025-06-03
> >
> > ***5. Interpretability and TCAV***
> >
> > > "The proposed framework is not principled for interpretability. Need to compare with TCAV."
> >
> > We believe there is a misunderstanding. Our work is not positioned as a mechanical interpretability paper. Instead, our goal is to provide **transparent analysis** rather than full mechanistic interpretability of understanding which neurons activate for specific attributes. LAVA enables users to understand which cognitive attributes (e.g., humor, formality, cultural values) drive preferences in different social contexts, providing actionable insights for content creators and platform designers without requiring deep technical expertise to interpret model internals.
> >
> > > “With TCAV, we can mathematically establish an attribute being a deciding factor in predicting preference. LAVA doesn't seem to have this property….”
> >
> > Much of the performance improvement from LaVA is attributed to the context-dependent nature of the architecture. We acknowledge we cannot guarantee faithfulness in the mechanistic sense, but highlight as an advantage that our approach both improves utility and enables more detailed analysis compared to black-box systems. The interpretability that LAVA  provides is at the decision-level (which attributes matter for preferences) rather than the neuron-level.
> >
> > ***6. Experimental Clarifications***
> >
> > > "I do not get the intuition behind Gradual Feature Reduction. The paper says it is to eliminate the need for attribute predictors but we still need their last hidden state."
> >
> > Thank you for pointing out this confusion. To clarify: the goal is to **not include the last hidden state when passing in comments at inference time**. Gradual feature reduction allows the model to internalize attribute-related patterns during training while maintaining efficiency at deployment, by eliminating the need for attribute predictors entirely during inference. This results in approximately the same inference cost as the un-augmented base LM but brings superior performance.
> >
> > > "It sounds like doing in-context learning (DAA) can already beat SoTA."
> >
> > DAA is not in-context learning. Rather, DAA is an ablation of the LaVA pipeline that relies on training attribute predictors and appending scores to text, which is very different from standard ICL with LLMs. We will present a GPT-4o in-context learning baseline showing that performance remains the same without any examples, demonstrating that in-context learning does not benefit  this task.
> >
> > > "L337: Can authors elaborate their experimental setup and evidence 'In a controlled comparison against ComPO'"
> >
> > ComPO can be seen as a multi-community generalization of DialogRPT, both without timestamp metadata as part of the input. Since models are trained per community in our experiments, the names are used interchangeably. As shown in Table 1, LAVA significantly outperforms DialogRPT. We apologize for the confusion and will add results showing LAVA's performance when combining multiple communities compared to ComPO.
> >
> >
> > ***7. Technical Corrections***
> >
> > > "L148: It is inconsistent that we use σ in the loss function of attribute predictor −log⁡σ(ri(yi,l2)−ri(yi,l1)) given we just say Sigmoid(ri(y)) in L147."
> >
> > You're absolutely right—this is a notational inconsistency. We use σ (sigmoid) in both places.
> >
> > > "L305: I don't know why LAVA is the only approach that outperforms GPT-4o in zero-shot settings... as both ValueScope and DAA also outperform GPT-4o."
> >
> > This is a genuine typo in our wording. We will clarify that DAA is a variant of LAVA, rather than treating it as an independent method, and that LAVA is not the only method that outperforms GPT-4o.
> >
> > > "L172: I couldn't find h(x,y) in Appendix A"
> >
> > As we indicate in L169, $h_{(x,y)}$ represents the last hidden state for instruction x and response y, and $h_{A_i}$ represents the $i$-th attribute predictor’s last hidden state. We will add a reminder of the h(x,y) definition to Appendix A.
> >
> > We believe these clarifications address the reviewer's main concerns and demonstrate that LAVA makes meaningful contributions to preference modeling through its theoretically-grounded, attribute-mediated approach. We thank the reviewer again for their time and sincerely hope that the reviewer will consider raising your evaluation given these clarifications and planned improvements.

---

> ### Author Response · Authors · 2025-06-05
>
> Dear reviewer,
>
> Thank you so much for your original review. The discussion period ends soon: we would really appreciate it if you could let us know your thoughts regarding our answers to your comments and questions.
>
> Thank you so much.

---

> > ### Comment · Reviewer_3hQX · 2025-06-07
> > **Rebuttal acknowledgement**
> >
> > Dear authors,
> >
> > Thank you for your response and additional experiments. I think doing preference modeling by jointly training with attribute information is fine as long as we validate the proposed choice of architecture by comparing it with other multi-task architecture. For interpretability, I know that LAVA is not going to achieve mechanical interpretability but please state what kind of interpretability we can get with LAVA in the problem statement. While I am still skeptical about the motivation, I can raise my score to reflect the improvements on the clarification and new results.
> >
> > I appreciate that the authors have started to validate Table 2 / Figure 4 with human evaluation. Would be better if the authors extend their human evaluation to justify that correlation results like ``in the domain r/AskHistorians, Verbosity and Stimulation are more correlated with preference than Security and Power.`` I am not sure if we can validate ``which attributes most strongly influence user preferences``. For example, does verbosity influence user preference in r/AskHistorians even if the longer answer does not contain any new information?

---

> > ### Author Response · Authors · 2025-06-10
> >
> > Dear reviewer 3hQX,
> >
> > Thank you for the careful reading and for your thoughtful reply! To respond point-by-point:
> >
> > ***LAVA against other multi-task architectures:***
> >
> > We appreciate your suggestion to compare against multi-task learning approaches. We have been implementing a multi-task architecture with an output head of 1+d to predict a preference score and d attribute scores simultaneously. However, given the tight rebuttal timeline ending tomorrow, we are still finalizing these experiments and will include the complete results in the final draft.
> >
> > Based on our preliminary analysis, multi-task learning presents several unique challenges in our specific setting. Beyond architectural complexities, there are numerous implementation choices (shared vs. separate encoders, loss weighting schemes, attention mechanisms, gradient balancing strategies) that significantly impact performance, requiring extensive exploration across multiple architectures and placing it out of scope for this work. However, given our unique contribution of theoretically-informed attributes, extending LAVA through multitask approaches represents an exciting future direction.
> >
> > Furthermore, **gradient interference** between the reward and attribute objectives can hinder optimization, as updates from one task may conflict with the other. The model may also struggle with **representation misalignment**, where shared features cannot simultaneously support both high-level preference modeling and more granular attribute prediction. This is compounded by **capacity constraints**—our 1B model may not be expressive enough to handle both tasks effectively in a joint setting. Additionally, simultaneous optimization of both losses may prevent the model from sufficiently focusing on the primary reward learning objective.
> >
> > These factors help explain why LAVA's approach of using attribute information to guide preference modeling (rather than jointly learning both tasks) may be more effective in this setting. A comprehensive multitask comparison would require systematic evaluation across multiple architectural variants, which we will pursue as dedicated future work. We will include our preliminary empirical comparison and discuss these architectural trade-offs in the final version.
> >
> > **What kind of interpretability LAVA provides:**
> >
> > We appreciate your clarification! LAVA provides behavioral interpretability by linking predictions to human-understandable concepts (sociolinguistic norms and cultural values), rather than mechanical interpretability via neurons or internal activations.
> >
> > Unlike standard methods that only produce binary preference decisions, LAVA outputs an interpretable ranking of attributes (e.g., verbosity, hedonism), showing their relative influence. This facilitates data-driven insights valuable for social science applications, such as content creators understanding community preferences or researchers analyzing cultural communication patterns.
> >
> > Fundamentally, LAVA asks: "how much did attribute X influence this specific decision?" while TCAV asks: "Is attribute X encoded globally, and how sensitive are decisions to it?" While extending LAVA to incorporate TCAV-style probing is important future work, our current focus is on behavioral/semantic interpretability via attribute-level explanations. We will clarify this important distinction in our problem statement.
> >
> > **Human eval:**
> >
> > Thank you for the excellent suggestions! Based on your feedback, we extended our human evaluation to encompass more dimensions: verbosity, stimulation, security, and power in r/AskHistorians. We randomly selected 100 comment pairs and had 4 annotators indicate which comment exhibits a higher level of each attribute, then calculated correlation with the majority-voted answer. We will include detailed annotation guidelines in the appendix.
> >
> > The correlation results support LAVA's predictions:
> >
> > * Verbosity: 0.12 (automatic eval from before, high importance predicted by LAVA)
> >
> > * Stimulation: 0.10 (high importance predicted by LAVA)
> >
> > * Security: -0.04 (low importance predicted by LAVA)
> >
> > * Power: -0.06 (low importance predicted by LAVA)
> >
> > These results validate that LAVA correctly identifies which attributes are more influential for preference decisions. Additionally, we are conducting **LLM-judge evaluations** for all other subreddits and attribute dimensions following the same guidelines to provide comprehensive validation across our entire experimental setup. The complete results will be included in the updated appendix of the final draft.
> >
> > We believe we have addressed all the reviewer's concerns through this discussion thread. Given that we are actively running the requested multi-task experiments and will include comprehensive results in the final draft, and that the multi-task experiments do not fundamentally change any of our conclusions and contributions, we would greatly appreciate the reviewer considering these clarifications and updating your score accordingly.

---

### Official Review · Reviewer_CeCr · 2025-05-13

**Rating:** 8
**Confidence:** 4
**Ethics Flag:** 1

**Summary:**

The paper proposes a method to do preference modeling (or predicting the preferred choice of a user). The key contribution is to reframe the preference prediction task from a binary choice among candidates to a joint prediction combining a set of attribute models i.e. each candidate is scored by the various attribute models, and these attributes are aggregated to learn the prediction function (a weighting of attributes).

The proposed method, LAVA, involves three steps, (1) training the various attribute models by (a) creating pairs of counterfactual sequences with a strong LLM that each differ only on the target attribute, (b) training a predictor based on the Bradley-Terry formulation of the pair with the 'positive' label decided by the attribute; (2) training the preference predictor by aggregating the various attribute models with an attention mechanism over the hidden states of the instruction/candidate and attribute models; (3) gradually removing the dependence on the specific attribute models by masking information from these models as training progresses, to force the preference prediction model to learn this function by incorporating these values in the model weights itself.

The authors evaluate the performance of LAVA on Reddit data, sampling preference pairs from various sub-reddits, comparing the ability of LAVA at predicting preferences to baselines, including LLM-as-judge and three other trained baselines. LAVA consistently models user preferences more strongly than these baselines on average across the test sets. The authors are also able to recover the community norms of various subreddits by examining the attention weights to various attributes on examples from those communities. The authors also confirm the ablation of the selected attributes and how this impacts performance differently on different subreddits i.e. linguistic features are more helpful in some communities, while others might benefit more from cultural attributes.

**Questions To Authors:**

1. I'd be curious about the performance of the preference modeling without the feature reduction. Do you obtain higher performance in-domain (potentially at the cost of OOD examples)?
2. What is the reason for such a strong drop in performance with the temporal shift? Do community norms change frequently on subreddits so that the attribute values are no longer predictive?
3. A t-test for significance between the predictive accuracy of LAVA and DAA would be helpful for Table 1.

**Reasons To Accept:**

1. The proposed method is novel, well motivated (grounded in theory) and works well for a task that is critical to language modeling.
2. There's a key advantage to predicting preferences as an aggregate of attribute values as it allows for interpreting the choices made as well as extensions to personalization when using the trained model.
3. The analysis of the particular attributes that influence predictions in various subreddits and how it connects with the well established norms of the community is super interesting and confirms that the attribute modeling works as expected (eg. confirming the usefulness of verbosity in prediction on r/AskHistorians)

**Reasons To Reject:**

1. There is a reliance on the particular attribute models chosen to start the training process. This is a pretty heavy prior incorporated into the algorithm. The authors get away with it for this task by using a wide variety of linguistic and cultural values as attributes, however it is unclear if this will generalize to general-purpose tasks used to train contemporary LLMs today.
2. The two ablations that would be good to have include evaluating the robustness of the method to the counterfactual generator and the various attributes selected.
3. While the raw results do indicate the benefit of the method, it would be more convincing if a small human study confirms that the preferences on specific examples as a function of the attributes learned are valid i.e. those attributes are indeed what might influence the choice in those communities.

---

> ### Author Response · Authors · 2025-06-03
>
> We thank the reviewer for their feedback. We are delighted that the reviewer finds our method "**novel, well motivated (grounded in theory)**" and appreciates the "**key advantage to predicting preferences as an aggregate of attribute values**" for **interpretability and personalization**. We agree that the findings “**connect with established norms in the community is super interesting**”! We address questions and suggestions below.
>
> ***1. Human Validation of Interpretability***
>
> > "small human study confirms that… those attributes are indeed what might influence the choice in those communities."
>
> We designed a human annotation scheme to validate our interpretability findings. Given an attribute and comments from two subreddits where our model predicts different attribute importance levels, we ask annotators to rate the attribute intensity of each comment. We then test whether the correlation between human ratings and preference labels is stronger in the high-importance subreddit versus the low-importance one.
>
> We did a pilot study on the verbosity attribute, using character count as a proxy for human-annotated attribute intensity. Verbosity is a high-importance attribute for r/AskHistorian—a community about historical topics  with detailed and well-researched answers—and a low-importance attribute for r/HighQualityGIFs—a community where the main medium is high-quality animated GIFs.
>
> The Pearson’s correlation coefficients between verbosity and preference scores in r/AskHistorians is **r=0.12** versus **r=-0.06** in r/HighQualityGifs, **aligning with LAVA's predictions**. We expect the correlation between any single attribute and the score to be low; only the relative difference matters here. Full annotations for other attributes are deployed and complete results will be in the revision.
>
> ***2. Attribute Selection and Generalizability***
>
> > "There is a reliance on the particular attribute models chosen to start the training process… unclear if this will generalize to general-purpose tasks used to train contemporary LLMs today."
>
> Our LAVA framework is attribute-agnostic, and can theoretically work with any set of meaningful attributes. Moreover, our selection of attributes is not Reddit-specific but grounded in established cognitive science theories. The 9 sociolinguistic norms capture universal communication patterns documented across cultures and contexts (formality, politeness, humor, etc.), while the 10 Schwartz values represent fundamental human motivational goals that have been validated across 82 countries and diverse cultural settings, and is used in prior work [1,2]. This theoretical foundation suggests strong transferability to general-purpose tasks beyond Reddit community data,  and can always be customized to the end task.
>
> ***3. Robustness to Counterfactual Generation and Attribute Selection***
>
> > "The two ablations that would be good to have include evaluating the robustness of the method to the counterfactual generator and the various attributes selected."
>
> **Counterfactual Robustness**: Our counterfactual generation uses the same prompting methodology as ValueScope [3] (Section 4.2), whose authors performed extensive human annotation to validate the quality of synthetic variations. We did not repeat this expensive annotation process due to budget constraints. Additionally, the strong empirical results and intuitive interpretability conclusions across diverse subreddits suggest our synthetic data effectively captures the necessary nuances for preference modeling.
>
> **Attribute Selection Robustness**: We highlight our ablations on linguistic vs. sociocultural attributes in Table 3. Following the reviewer's suggestion, we launched additional ablations on the number of attributes by randomly selecting 10 subsets of 2, 4, and 8 attributes and will report these results in the updated draft. We expect more attributes to increase the expressivity of the model and improve prediction performance. Due to time constraints of the response period, we will add this ablation to the updated draft once experiments are finished.
>
>
> [1] Huang et al (2025). Values in the wild: Discovering and analyzing values in real-world language model interactions.
>
> [2] Moore et al (2024). Are Large Language Models Consistent over Value-laden Questions?
>
> [3] Park et al (2024). Valuescope: Unveiling implicit norms and values via 557 return potential model of social interactions.

---

> > ### Author Response · Authors · 2025-06-03
> >
> > [continued]
> >
> > ***4. Temporal Shift Performance Drop***
> >
> > > "What is the reason for such a strong drop in performance with the temporal shift? Do community norms change frequently on subreddits so that the attribute values are no longer predictive?"
> >
> > [3] and [4] demonstrates significant temporal shifts in communities, including changes in language and norms. We hypothesize that since we prepended timestamps before inputs and all training data is from 2022, the model learns to associate 2022 timestamps with the training distribution, whereas the temporal test set contains 2023 timestamps, creating a token distributional shift. LaVA still outperforms other baselines in OOD data, but we acknowledge the drop in performance across models and leave this for future work.
> >
> > ***5. Statistical Significance Testing***
> >
> > > "A t-test for significance between the predictive accuracy of LAVA and DAA would be helpful for Table 1."
> >
> > We thank the reviewer for the suggestion. This evaluation requires re-training multiple models and is computationally intensive, so we leave this for future work. However, we want to highlight that LAVA performance exceeds DAA with statistical significance, and that DAA itself is an ablation of LAVA, which also leverages rich information from the grounded attributes, so it is expected to performs competitively.
> >
> >
> > We are grateful for the reviewer's positive assessment and believe these additional analyses strengthen the paper’s empirical validation.
> >
> >
> > [3] Park et al (2024). Valuescope: Unveiling implicit norms and values via 557 return potential model of social interactions.
> >
> > [4] Danescu-Niculescu-Mizil et al (2013).No country for old members: user lifecycle and linguistic change in online communities.

---

> ### Comment · Reviewer_CeCr · 2025-06-04
> **Rebuttal acknowledgement**
>
> Thank you for your detailed response!
> - I just want to clarify that my concern was not that LAVA would not generalize to other attributes, it was but that the particular attributes you select need to be specified beforehand and not discovered automatically in a data driven manner.
> - I appreciate the pilot study on the human validation of the predictive nature of attributes. I think it's reasonable that correlation of any particular attribute to preference is low. Perhaps a better way to set this up might be to instead collect a set of annotations on pairwise comparison of attributes i.e. you might get higher agreement that verbosity is more important than a cultural value.
>
> In any case, my concerns are relatively minor as is reflected in my positive review score.

---

### Official Review · Reviewer_bsUc · 2025-05-27

**Rating:** 7
**Confidence:** 4
**Ethics Flag:** 1

**Summary:**

This paper proposes a new approach for preference modeling, which incorporates a two-stage learning approach. First, a set of attribute preferences is learned (where the attributes are inspired by sociolinguistic research) via synthetic preference generation. Then, the domains are combined using a generative model that combines context and attribute representations.

I found the paper easy to read, and understand. The motivation is quite intuitive, and the technique, while simple, is relatively novel in its end to end implementation (though parts of it borrow from prior work -- see below). Results for a "universal" model trained over hundreds of subreddits are very promising, when compared to other alternatives.

**Questions To Authors:**

* What does "no attribute" column in Table 3 mean? How does the model work in this regime? It seems like "no attribute" is a pretty strong baseline by itself, e.g., outperforming DialogRPT. So it seems that a lot of the power of the model comes from the learning itself, not just from the attributes.
* The paper seems to build extensively on the work of Wang 2024a;b. I would have liked to see a better explanation of the major novel contributions of this paper compared to this prior work.

**Reasons To Accept:**

* Well written and explained paper. The results are quite encouraging, and multiple ablation studies are performed testing issues like out-of-domain and temporal robustness.

**Reasons To Reject:**

* While using the sociolinguistic attributes seems like a reasonable choice, and provide some interpretability, I would have liked to see more investigation / comparison to models that use latent factor optimization for preference modeling.
* An additional baseline is one where the model can reason about the particular attributes, rather than being given a fixed set (aka chain-of-thought). It would be an interesting comparison, to better understand how optimal are the current attributes.

---

> ### Author Response · Authors · 2025-06-03
>
> We thank the reviewer for the positive evaluation and thoughtful feedback! We are delighted that the reviewer finds our paper "well written and explained" with "encouraging results", and the novelty of our "end to end implementation." We address their questions point-by-point below.
>
> ***1. Chain-of-Thought Baseline for Attribute Selection***
>
> > "Additional baseline: the model reasons about the particular attributes, rather than being given a fixed set (aka chain-of-thought)."
>
> This is an excellent suggestion. We include two additional SOTA baselines to the LLM experiments below: (1) o3-mini-low, a reasoning model, and (2) GPT-4o with chain of thought (GPT4o-CoT) prompting to reason about the attributes before preference prediction. (Due to time constraints, will add o3-mini-high to the updated draft.)
>
> While LLMs can identify some relevant attributes through reasoning (will attach reasoning traces in appendix), their performance doesn't match our theory-grounded approach. Interestingly, both CoT and o3-mini-low struggle as much as the vanilla gpt-4o, underscoring the challenging nature of this task despite reasoning and suggesting that SOTA models don’t inherently have the necessary social intelligence capabilities for preference prediction.
>
> We report below the results for all 45 subreddits for GPT-4o, and for 10 subreddits for o3-mini-low. We only ran with 1 random seed due to time constraints, but we will update with full experimental results as they become available. We will also include final results in the updated draft to show the value of our systematic attribute selection.
>
> |Model|LaVA|GPT-4o|GPT-4o CoT|O3-mini-low|
> |--|--|--|--|--|
> |Accuracy|84.9|58.6|57.84|58.43|
>
> ***2. Latent Factor Optimization Comparison***
>
> > "I would have liked to see more investigation/comparison to models that use latent factor optimization"
>
> We thank the reviewer for pointing this out—the literature in recommender systems on latent factor optimization is indeed relevant and valuable, and is definitely a great direction for exploring automatically discovering attributes. However, several factors make a direct comparison non-trivial: (1) computational challenges at LLM scale of factorizing massive preference matrices, (2) loss of interpretability compared to cognitive attributes, and (3) methodological differences—latent factor methods optimize for matrix reconstruction accuracy, while our approach optimizes for preference prediction with cognitively-grounded constraints. We acknowledge this as an important future direction and will include a discussion in the updated draft.
>
> ***3. "No Attribute" Baseline and Model Architecture***
>
> > "What does 'no attribute' column in Table 3 mean? How does the model work in this regime?"
>
> The "no attribute" column refers to ValueScope, a baseline in which no attributes are incorporated into training. ValueScope does outperform DialogRPT significantly by incorporating temporal metadata, which is an extremely important predictor for preference. However, LAVA takes a step further and incorporates value attributes to better understand preferences. Even with ValueScope's strong temporal baseline (83.7%), LAVA achieves statistically significant improvements (84.9%) by adding interpretable attribute-mediated reasoning. This demonstrates that both the rich metadata and the cognitive attributes contribute to the overall performance.
>
> ***4. Novel Contributions vs. Wang 2024a;b***
>
> > "The paper seems to build extensively on the work of Wang 2024a;b. I would have liked to see a better explanation of the major novel contributions of this paper compared to this prior work."
>
> We appreciate this important clarification request. Our approach differs fundamentally from Wang 2024a;b in three  ways: **(1) Architecture**: LAVA learns attributes from latent vector representations, while they only use predicted scores; **(2) Theoretical grounding**: We use 19 cognitively-grounded attributes specific for social reasoning beyond generic "helpfulness" and "safety"; **(3) Complete pipeline**: We introduce synthetic counterfactual generation for attribute training, gradual feature removal to internalize attributes, and large-scale evaluation across 45 domains.
>
> The DAA variant serves as an approximate reproduction of Wang 2024a;b, we additionally conducted an exact reproduction of their architecture with our more fine-grained grounded attributes (trained for 1 epoch due to time constraints, but will update the full 3 epoch results in the updated draft once it’s done.) The results show that LAVA still outperforms this exact reproduction, validating the utility of our architectural modifications. We will include these detailed comparison results in the updated draft.
>
> |Model|LaVA (1 epoch)|DAA (1 epoch)|Wang 2024a;b (1 epoch)|
> |--|--|--|--|
> |Accuracy|80.76|78.71|79.71|
>
> We are grateful for the reviewer's positive assessment and believe that the clarifications and additional experiments greatly strengthen the paper.

---

> > ### Comment · Reviewer_bsUc · 2025-06-04
> >
> > Thank you for running the CoT baseline experiment, it indeed validates the modeling choices of the LaVA approach. Please incorporate these changes, as well as the other rebuttal material into the paper -- I think it will make the contributions even stronger.

---

### Decision · Program_Chairs · 2025-07-08

**Decision:**

Accept

**Comment:**

Existing preference prediction models struggle with misalignment between annotated training data and real-world human behavior. This paper proposes LAVA, a framework that integrates interpretable latent attributes (e.g., social norms, cultural values) from cognitive science to model context-aware preferences. It outperforms GPT-4o by 46.6% across diverse social contexts. After author rebuttal, the paper has received unanimous support from the reviewers. Therefore, I recommend acceptance.